# Cell circuits between leukemic cells and mesenchymal stem cells block lymphopoiesis by activating lymphotoxin beta receptor signaling

Xing Feng[1], Ruifeng Sun[1,2], Moonyoung Lee[3], Xinyue Chen[4], Shangqin Guo[4], Huimin Geng[5], Marcus Müschen[1,2], Jungmin Choi[3,6]*, Joao Pedro Pereira[1]*

[1]Department of Immunobiology and Yale Stem Cell Center, Yale University School of Medicine, New Haven, United States; [2]Center of Molecular and Cellular Oncology and Department of Immunobiology, Yale University, New Haven, United States; [3]Department of Biomedical Sciences, Korea University College of Medicine, Seoul, Republic of Korea; [4]Department of Cell Biology and Yale Stem Cell Center, Yale University, New Haven, United States; [5]Department of Laboratory Medicine, University of California, San Francisco, San Francisco, United States; [6]Department of Genetics, School of Medicine, Yale University, New Haven, United States

**Abstract** Acute lymphoblastic and myeloblastic leukemias (ALL and AML) have been known to modify the bone marrow microenvironment and disrupt non-malignant hematopoiesis. However, the molecular mechanisms driving these alterations remain poorly defined. Using mouse models of ALL and AML, here we show that leukemic cells turn off lymphopoiesis and erythropoiesis shortly after colonizing the bone marrow. ALL and AML cells express lymphotoxin α1β2 and activate lymphotoxin beta receptor (LTβR) signaling in mesenchymal stem cells (MSCs), which turns off IL7 production and prevents non-malignant lymphopoiesis. We show that the DNA damage response pathway and CXCR4 signaling promote lymphotoxin α1β2 expression in leukemic cells. Genetic or pharmacological disruption of LTβR signaling in MSCs restores lymphopoiesis but not erythropoiesis, reduces leukemic cell growth, and significantly extends the survival of transplant recipients. Similarly, CXCR4 blocking also prevents leukemia-induced IL7 downregulation and inhibits leukemia growth. These studies demonstrate that acute leukemias exploit physiological mechanisms governing hematopoietic output as a strategy for gaining competitive advantage.

*For correspondence:
jungminchoi@korea.ac.kr (JC);
joao.pereira@yale.edu (JPedroP)

## Editor's evaluation

This study investigates a novel pathway by which leukemic cells remodel the bone marrow niche to promote their expansion at the expense of normal hematopoiesis. Feng X, Pereira JP et al. convincingly demonstrate a positive feedback loop between leukemic cells and stromal cells mediated by lymphotoxin produced by cancer cells and its receptor expressed by bone marrow stromal cells. The authors provide compelling evidence suggesting that this pathway disrupts normal blood production and provides a competitive advantage to leukemic cells.

## Introduction

Blood cell production is a tightly regulated process important for organismal homeostasis. All blood cells develop from a dedicated hematopoietic stem cell (HSC) that colonizes specialized niches in the

bone marrow (BM) formed predominantly by mesenchymal stem cells (MSCs) and endothelial cells (ECs) (*Morrison and Scadden, 2014*; *Pinho and Frenette, 2019*; *Sugiyama et al., 2019*). Within these niches, HSCs and hematopoietic progenitors receive critical signals for long-term HSC maintenance and for differentiation into lymphoid, myeloid, and erythroid lineages (*Sugiyama et al., 2019*; *Miao et al., 2020*). However, most hematopoietic cytokines act in a short-range manner, and thus hematopoietic stem and progenitor cells rely on localization cues such as CXCL12 for accessing growth factors produced by MSCs and ECs (*Noda et al., 2011*; *Tzeng et al., 2011*; *Ding and Morrison, 2013*; *Greenbaum et al., 2013*; *Cordeiro Gomes et al., 2016*).

While HSCs and uncommitted hematopoietic progenitors are critically dependent on stem cell factor (SCF, encoded by *Kitl*), committed progenitors require lineage-specific signals, such as IL7 for lymphocytes, IL15 for NK cells, or M-CSF for monocytes and macrophages. Importantly, most hematopoietic cytokines are produced by MSCs and by a subset of ECs in the BM (*Miao et al., 2020*; *Noda et al., 2011*; *Cordeiro Gomes et al., 2016*; *Ding et al., 2012*; *Baryawno et al., 2019*; *Comazzetto et al., 2019*; *Tikhonova et al., 2019*). The production of hematopoietic cytokines and chemokines by MSCs and ECs is relatively stable during homeostasis but can change significantly under certain perturbations. For example, systemic inflammation caused by infections enforces the downregulation of multiple hematopoietic cytokines and CXCL12 in the BM (*Ueda et al., 2004*; *Ueda et al., 2005*; *Manz and Boettcher, 2014*). Likewise, acute lymphoblastic and myeloblastic leukemias (ALL and AML) also promote the downregulation of multiple cytokines and CXCL12 produced by MSCs and ECs (*Baryawno et al., 2019*; *Hanoun et al., 2014*; *Fistonich et al., 2018*; *Zehentmeier and Pereira, 2019*). During systemic infection, the coordinated downregulation of certain cytokines (e.g. IL7) and CXCL12 causes a temporary pause in lymphopoiesis that is necessary for an emergent production of short-lived neutrophils and monocytes (*Manz and Boettcher, 2014*). In leukemic states, however, the mechanism(s) promoting cytokine and chemokine downregulation are not well defined, and neither is known if these changes are protective or harmful for the host.

In humans and in mouse models of B-ALL, leukemic cells use CXCR4 to home to the BM (*Juarez et al., 2007*; *Colmone et al., 2008*; *van den Berk et al., 2014*). However, B-ALL cells do not distribute randomly and seem to reside and proliferate in certain perivascular niches (*Colmone et al., 2008*; *Sipkins et al., 2005*). Importantly, CXCL12 production is measurably reduced exclusively in BM niches colonized by B-ALL cells (*Colmone et al., 2008*; *van den Berk et al., 2014*). Furthermore, intact CXCR4 signaling presumably in B-ALL cells is required for downregulation of CXCL12 expression in BM niche cells (*van den Berk et al., 2014*). The fact that CXCR4 expression levels in B-ALL cells inversely correlate with patient outcome suggests that B-ALL-induced changes in the BM microenvironment may favor leukemia progression (*van den Berk et al., 2014*; *Cancilla et al., 2020*).

The BM microenvironment has also been reported to be severely affected in AML patients and in mouse models of AML. Of note, hematopoietic cytokines and chemokines are significantly downregulated along with the re-programing of MSC and EC transcriptomes (*Baryawno et al., 2019*; *Hanoun et al., 2014*; *Chandran et al., 2015*; *Geyh et al., 2016*). Although no specific mechanisms have been identified for explaining how AML cells dysregulate MSCs and ECs, some evidence suggests that this may be mediated by direct AML-niche cell interactions (*Hérault et al., 2017*). Thus, a model emerges where leukemia cells attracted to CXCL12-producing BM niches physically interact and re-program MSCs and ECs to reduce CXCL12 levels, possibly reduce hematopoietic output, and in this way favor leukemic cell expansion. However, the molecular mechanisms utilized by leukemia for MSC and EC re-programing and for reducing non-malignant hematopoiesis remain poorly defined.

In this study, we show that ALL and AML cells preferentially turn off lymphopoiesis and erythropoiesis shortly after seeding the BM. We demonstrate that both B-ALL and AML cells express LTα1β2, the membrane-bound ligand of lymphotoxin beta receptor (LTβR), which enforces IL7 downregulation in LTβR-expressing MSCs. Genetic or pharmacological blockade of LTβR signaling in MSCs restores lymphopoiesis but not erythropoiesis at the onset of leukemia, which in turn reduces leukemic cell growth and extends survival of transplant recipients. These studies demonstrate that leukemic cells exploit molecular mechanisms that confer flexibility in blood cell production to suppress normal hematopoiesis.

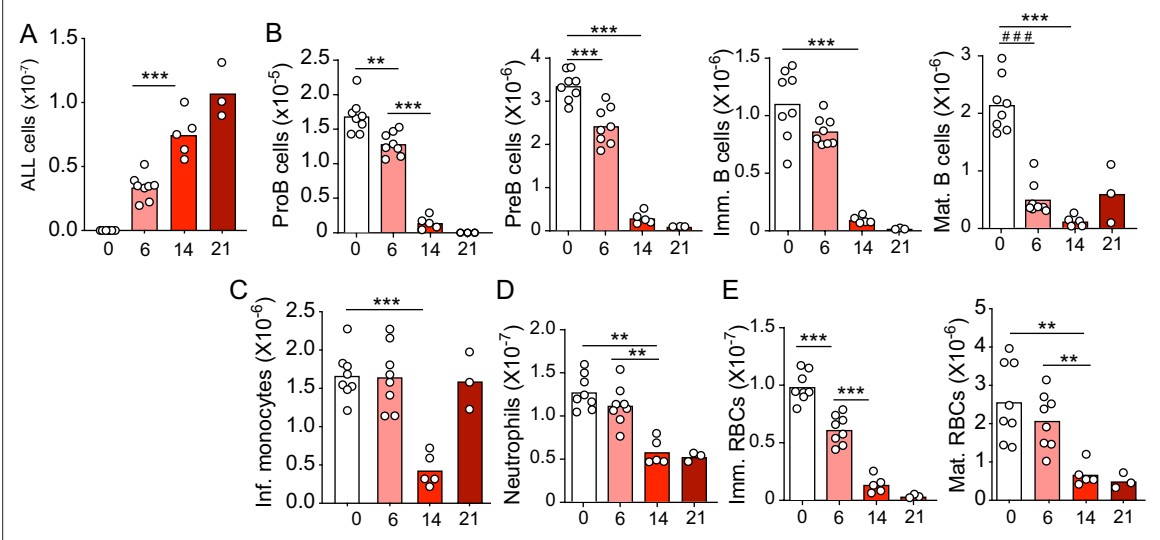

**Figure 1.** Kinetics of B-ALL growth and impact on hematopoiesis. (**A**) B-ALL number. (**B**) Number of non-malignant developing B cell subsets. (**C**) Inflammatory monocytes. (**D**) Neutrophils. (**E**) Immature (Ter119+CD71+) and mature (Ter119+CD71-) red blood cells. Data in all panels show bone marrow cell numbers obtained from wild-type (WT) mice transplanted with 3×10⁶ BCR-ABL-expressing B-ALL cells. In all panels, X-axis indicates time (days) after B-ALL transplantation. Bars indicate mean, circles depict individual mice. Data are representative of two independent experiments. **p<0.005; ***p<0.0005 unpaired, two-sided, Student's *t* test. ###p<0.0005 Mann–Whitney test. ALL, acute lymphoblastic leukemia.

# Results

## ALL inhibits non-leukemic hematopoiesis

Although leukemia alters BM niches, whether these changes directly affect hematopoietic cell production has not been carefully studied. To determine if and which hematopoietic cell lineages are affected by leukemia, we transplanted 3 million pre-B-cell precursor ALL cells expressing the BCR-ABL1 oncogene (from here on referred as ALL; BCR-ABL1 reported by YFP expression) into non-irradiated C57Bl6/J mice and analyzed its impact on lymphoid, myeloid, and erythroid cell production over time. As expected, ALL cells expanded rapidly in BM (*Figure 1A*). Conversely, non-leukemic developing B cells and mature recirculating B cells declined sharply 2 weeks after ALL transplantation (*Figure 1B*). Monocyte numbers reduced between the first and second weeks by four- to fivefold, but cell numbers recovered to normal levels at 3 weeks (*Figure 1C*). This contrasted with a moderate twofold decline in neutrophil numbers at 2 weeks that remained stable until 3 weeks (*Figure 1D*). Changes in immature erythrocytes (Ter119+CD71+ cells) were similar to the reductions seen in B cell progenitors: erythroid cells progressively reduced at 6, 14, and 21 days after ALL transplantation, reaching >10-fold reductions at 3 weeks (*Figure 1E*). In summary, ALL expansion induces a strong decline in lymphopoiesis and erythropoiesis while their impact on myeloid cell production is modest.

## ALL induces LTβR signaling in MSCs, downregulates *Il7* expression, and modulates lymphopoiesis

In previous studies we noted that transplanted ALL cells and Artemis-deficient (pre-leukemic) pre-B cells led to IL7 and CXCL12 downregulation in MSCs (*Fistonich et al., 2018*), which could explain the negative impact of ALL in non-malignant lymphopoiesis. While the mechanism(s) responsible for IL7 and CXCL12 downregulation remained undefined, earlier studies suggested a role for LTβR signaling in BM stromal cells in development of some lymphoid lineages (*Wu et al., 2001*; *Kim et al., 2014*). In recent studies, we found that MSCs express LTβR and that LTβR signaling controls IL7 expression in vivo (*Zehentmeier et al., 2022*). Furthermore, when mRNA levels of *LTB* were analyzed in pediatric samples of B-ALL (Children's Oncology Group Study 9906 for High-Risk Pediatric ALL) and associated with clinical outcome at the time of diagnosis, we noted an inverse correlation between *LTB* transcript abundance and relapse-free survival that reached statistical significance (*Figure 2—figure supplement 1A*). These observations led us to hypothesize that leukemic cells express LTβR ligands

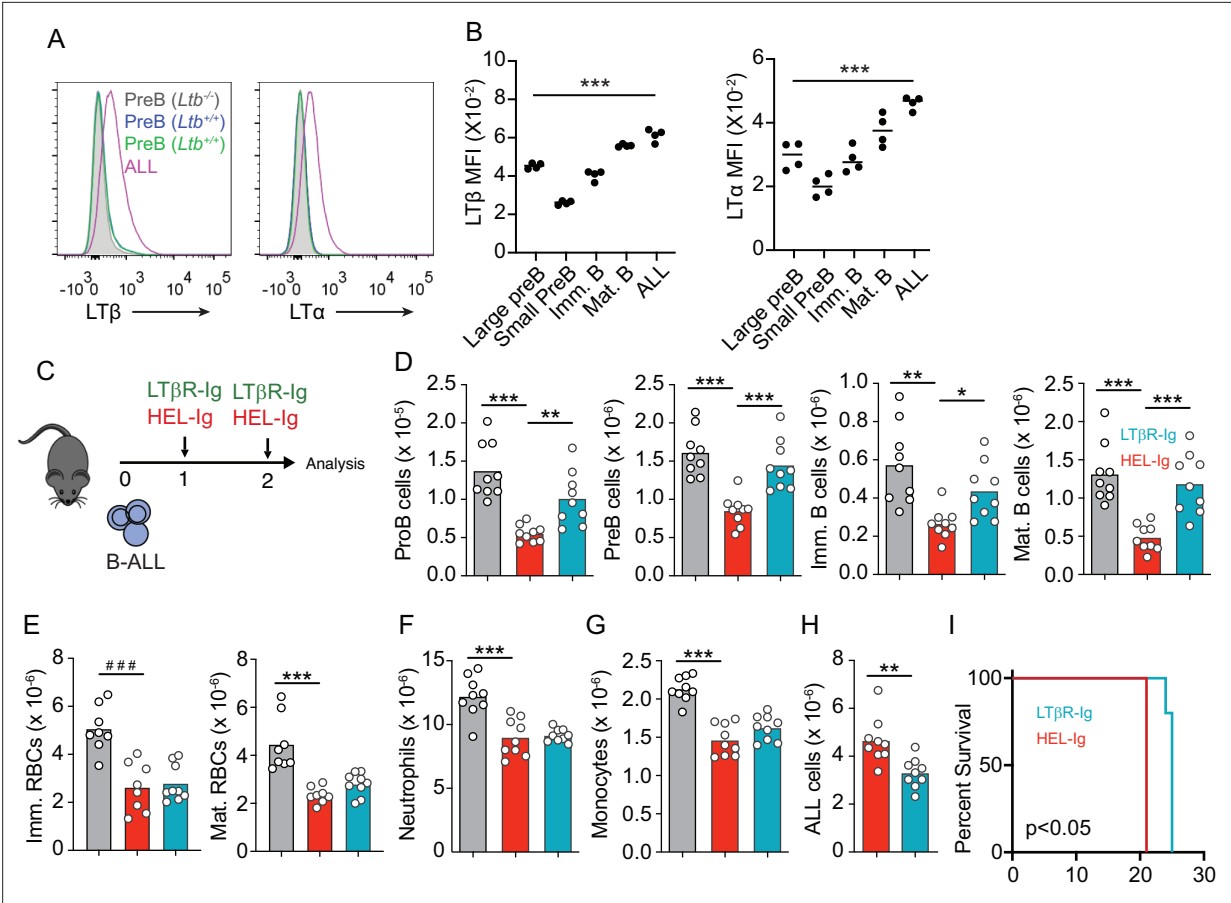

**Figure 2.** Lymphotoxin α1β2 expression in B-ALL cells and therapeutic effect of lymphotoxin beta receptor (LTβR) blocking. (**A**) Histograms of LTα and LTβ expression in B-ALL cells and in pre-B cells. Purple, B-ALL; green, non-malignant pre-B cells (CD19+CD93+IgM-cKit-) in bone marrow (BM) of wild-type (WT) mice transplanted with B-ALL cells; blue, non-malignant pre-B cells in BM of WT mice (no B-ALL); filled gray, non-malignant *Ltb*-deficient pre-B cells in BM of *Ltb*⁻/⁻ mice. (**B**) LTα and LTβ mean fluorescence intensity (MFI) in developing B cells and ALLs isolated from BM of ALL transplanted mice. (**C**) Experimental design of data described in panels D–H. (**D**) Number of non-malignant developing B cell subsets in BM. (**E**) Immature and mature erythrocyte number. (**F**) Neutrophils. (**G**) Monocytes. (**H**) B-ALL number. Data in panels D–H show BM numbers from WT mice transplanted with 3×10⁶ BCR-ABL-expressing B-ALL cells and treated with HEL-Ig or LTβR-Ig (150 μg/mouse) on day 0 and day 5; mice were analyzed on day 8 post ALL transplantation. (**I**) Frequency of mouse survival after B-ALL transplantation following pre-treatment with either HEL-Ig or LTβR-Ig (n=5 per group). Mice were treated with HEL-Ig or LTβR-Ig (150 μg/mouse) every 5 days until endpoint. Bars indicate mean, circles depict individual mice. Data are representative of two independent experiments. *p<0.05; **p<0.005; ***p<0.0005 unpaired, two-sided, Student's *t* test. ###p<0.0005 Mann–Whitney test.

The online version of this article includes the following figure supplement(s) for figure 2:

**Figure supplement 1.** Relationship between lymphotoxin α1β2 abundance and acute lymphoblastic leukemia (ALL) lethality.

and induce LTβR signaling in MSCs in vivo. In mice, BCR-ABL1 expressing pre-B ALL cells also express higher LTα/LTβ amounts than non-leukemic pre-B, immature, and mature B cells (***Figure 2A and B***). The presence of ALL cells in the BM environment did not change LTα and LTβ expression on non-leukemic host pre-B cells (***Figure 2A***). Importantly, when mouse ALL cells were engineered to over-express LTα/LTβ, these ALLs induced stronger IL7 downregulation in BM MSCs and were lethal more quickly than empty vector transduced ALL cells (***Figure 2—figure supplement 1B–E***). Combined, these studies suggest a pathogenic role for the LTβR pathway in leukemia progression.

To test if LTβR signaling impacts ALL growth and non-malignant hematopoiesis, we transplanted 3 million ALL cells into WT syngeneic recipient mice (C57BL6/J) treated weekly with a soluble LTβR-Ig decoy (a fusion between LTβR ectodomain and the Fc domain of a mouse IgG1 recognizing Hen Egg Lysozyme) or with control Hel-Ig. Transplanted ALL cells reduced lymphopoiesis significantly, which was reverted with LTβR-Ig treatment (***Figure 2C and D***). In contrast, LTβR signaling blockade did not restore erythropoiesis or myelopoiesis (***Figure 2E–G***). Importantly, ALL growth was significantly

reduced at 2 weeks (*Figure 2H*), which reflected in a small but significant extension of mouse survival (*Figure 2I*).

To gain further insight into the mechanisms used by ALL cells for reducing non-malignant hematopoiesis, we analyzed the MSC transcriptome in homeostasis, during ALL expansion, and in mice with ALL but treated with LTβR-Ig. To identify gene expression differences between the three groups, we performed principal component analyses (PCA) on the transcriptome datasets from three to four independent replicates. The first two principal components (PC1 and PC2) represent the main axes of variation within these datasets and explained 46% and 17% of variation, respectively. Samples from control and ALL groups separated by PC1, and within ALL cohorts, samples from LTβR-Ig versus Hel-Ig treated ALL also segregated from each other, thus indicating major transcriptional changes induced by ALL growth in vivo, of which a significant fraction was sensitive to LTβR blocking (*Figure 3A*). Unsupervised clustering of the top 1000 most variable genes also independently segregated the three groups (*Figure 3B*). Comparisons between control and ALL treated with Hel-Ig samples revealed 322 differentially expressed genes (DEGs; Padj <0.05, |log2FC|>1), of which 74 were downregulated and 248 were upregulated in MSCs of control mice (*Supplementary file 1*). Comparisons between the ALL groups (Hel-Ig versus LTβR-Ig) revealed 226 DEGs of which 149 were upregulated and 77 were downregulated in MSCs of mice with ALL and treated with Hel-Ig (*Supplementary file 2*). Gene set enrichment analyses revealed a strong inflammatory gene signature induced by ALL with a strong statistical significance in interferon α- and γ-induced genes, complement, and cytokines IL2, IL6, and TNFα signaling (*Figure 3C*). Of note, LTβR blocking further increased the interferon stimulated gene signature, while it reduced the expression of genes associated with TNFα signaling (*Figure 3C*), consistent with the fact that LTβR is a TNF superfamily member that activates canonical and non-canonical nuclear factor kappa-binding transcription factors (NFκB) (*Norris and Ware, 2007*). Importantly, of the several hematopoietic cytokines expressed by MSCs, KITL, IL7, IGF1, and CSF1 were significantly downregulated by ALL cells (*Figure 3D*). These transcriptional changes in MSCs were similar to those described in mice with acute myeloid leukemia (*Baryawno et al., 2019*). However, of these hematopoietic cytokines, only IL7 downregulation was blocked by LTβR-Ig treatment (*Figure 3D*). Furthermore, blocking other NFκB-inducing cytokines, such as TNFα and IL1β, did not prevent IL7 downregulation nor did it rescue non-malignant lymphopoiesis or myelopoiesis (*Figure 3—figure supplement 1A–C*) and did not impact ALL expansion in vivo (*Figure 3—figure supplement 1D*). Even though ALL cells promoted an interferon-induced gene expression signature in MSCs (*Figure 3C*), blocking IFNα or IFNγ signaling did not rescue IL7 downregulation and non-malignant hematopoiesis, nor did it reduce ALL growth in vivo (*Figure 3—figure supplement 1E–I*). Combined, these results show a major impact of ALL expansion in the MSC transcriptome, with a large fraction of DEGs being sensitive to LTβR blocking.

To test if LTβR signaling in MSCs impacts ALL growth, non-malignant hematopoiesis, and mouse survival, we transplanted ALL cells into mice conditionally deficient in *Ltbr* in MSCs (*Ltbr*$^{fl/fl}$; *Lepr*$^{Cre/+}$ mice, from here on referred as LTβRΔ) that also report *Il7* transcription via GFP expression (*Il7*$^{GFP/+}$). We ruled out a role for LTβR signaling in MSCs in promoting ALL homing to the BM by transplanting 3×10$^6$ ALL cells into control or LTβRΔ mice (*Figure 4A*), in agreement with prior studies showing that LTβR signaling in MSCs does not control CXCL12 expression under homeostatic conditions (*Zehentmeier et al., 2022*). Transplanted ALL cells induced IL7 downregulation in control mice (WT, *Lepr*$^{+/+}$; *Ltbr*$^{fl/fl}$) but not in LTβRΔ mice (*Figure 4B*), as expected (*Figure 3D*). These changes in IL7 production corresponded with reduced lymphopoiesis in WT mice whereas lymphopoiesis was largely unaffected in LTβRΔ mice (*Figure 4C–E*). In contrast, ALL-induced reductions in myeloid and erythroid lineages were largely independent of LTβR signaling in MSCs (*Figure 4G–J*). The inability to induce LTβR signaling in MSCs also impacted ALL growth in vivo (*Figure 4F and K*) such that it extended mouse survival by approximately 1 week (*Figure 4L*). To further test if ALL cells directly induce LTβR signaling in MSCs, we generated ALL cells genetically deficient in *Ltb* (*Figure 4M*). Indeed, *Ltb*-deficient ALL cells were unable to induce IL7 downregulation in MSCs and to block non-malignant lymphopoiesis (*Figure 4N and O*). Furthermore, *Ltb*-deficient ALL cells proliferated significantly less than *Ltb*-sufficient ALL cells (*Figure 4P*), which extended mouse survival significantly (*Figure 4Q*). Finally, to account for reduced *Ltb*-deficient ALL growth in vivo, we measured changes in *Il7* expression in mice transplanted with 3×10$^6$ *Ltb*$^{+/+}$ ALLs, 3×10$^6$ *Ltb*$^{-/-}$ ALLs, and 9×10$^6$ *Ltb*$^{-/-}$ ALLs (3×*Ltb*$^{-/-}$). Importantly, IL7 expression was unchanged even in mice transplanted with threefold higher number of *Ltb*-deficient

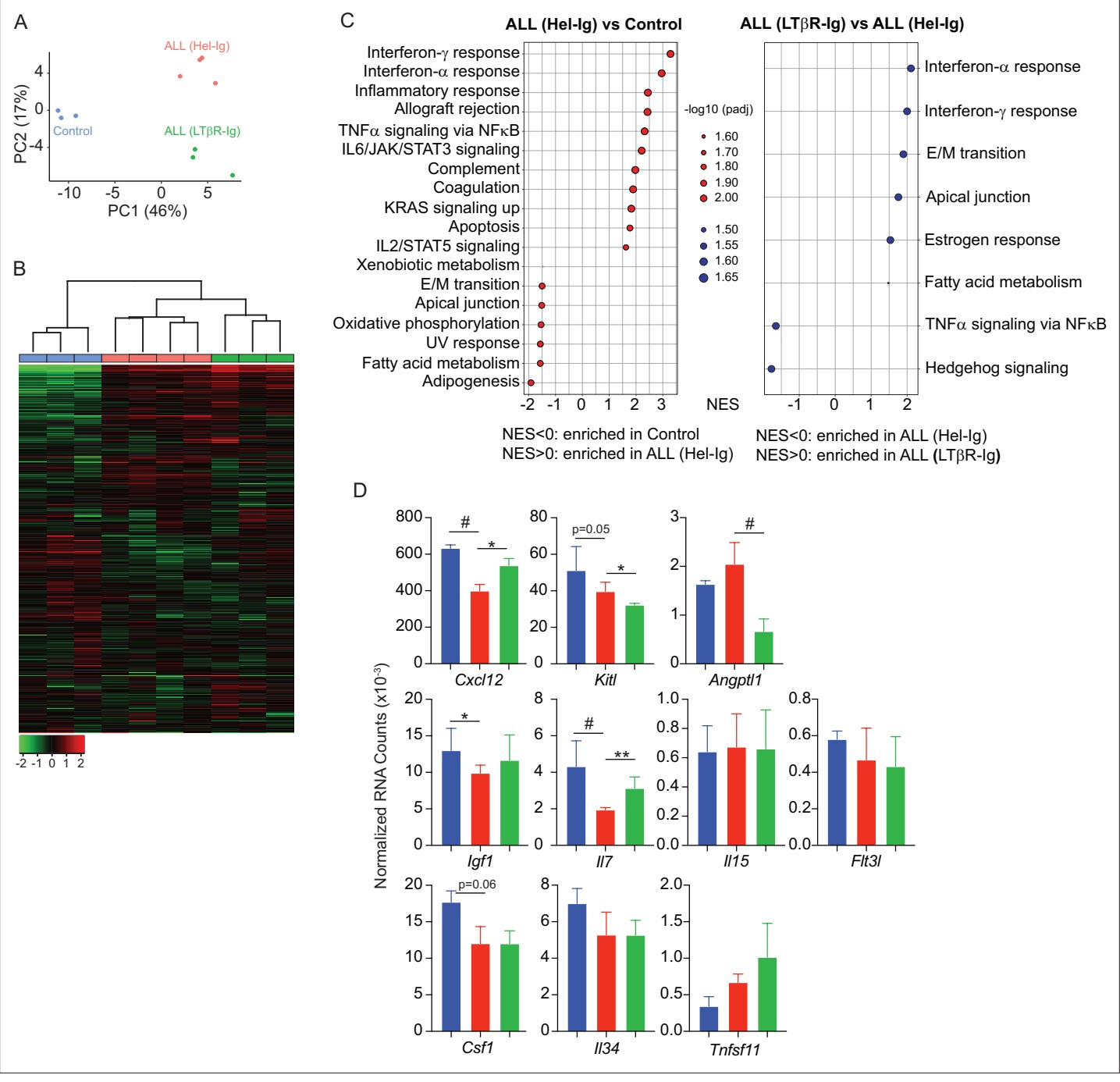

**Figure 3.** Lymphotoxin beta receptor (LTβR)-dependent and -independent transcriptomic changes in mesenchymal stem cells (MSCs) induced by B-ALL. (**A**) Principal component analysis (PCA) distribution plot. (**B**) Unsupervised hierarchical clustering and heatmap representation of top 1000 differentially expressed genes. (**C**) GSEA-KEGG pathway alterations in MSCs. (**D**) Hematopoietic cytokines and chemokine mRNA expression. Data in all panels were generated from analyses of MSC bulk RNA sequencing. *p<0.05; **p<0.005; unpaired, two-sided, Student's *t* test. ALL, acute lymphoblastic leukemia.

The online version of this article includes the following figure supplement(s) for figure 3:

**Figure supplement 1.** Effect of inflammatory cytokines in acute lymphoblastic leukemia (ALL) growth.

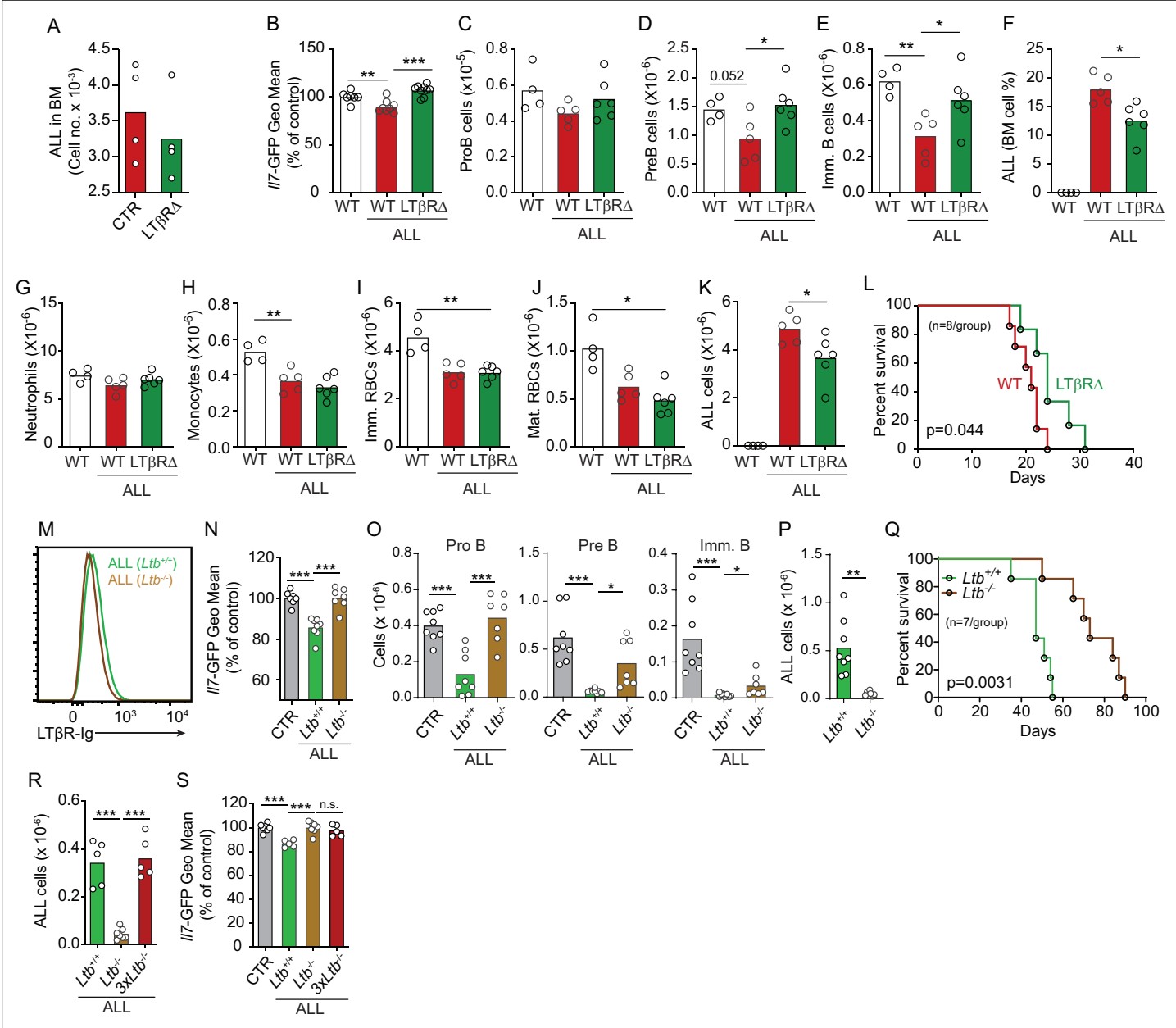

**Figure 4.** Effects of mesenchymal stem cell (MSC)-intrinsic lymphotoxin beta receptor (LTβR) signaling in lymphopoiesis and B-ALL growth.
(**A**) Measurements of ALL homing to the bone marrow (BM): 3×10⁶ BCR-ABL ALLs were transferred into control (red) or LTβRΔ (green) mice and allowed to home into the BM for 24 hr. (**B**) *Il7*-GFP expression in MSCs. (**C–E**) Number of non-malignant developing B cell subsets. (**C**) ProB cells. (**D**) Pre-B cells. (**E**) Immature B cells. (**F**) B-ALL frequency in BM. (**G–K**) Myeloid and erythroid cell numbers in BM. (**G**) Neutrophils. (**H**) Monocytes. (**I**) Immature RBCs. (**J**) Mature RBCs. (**K**) Total ALL number. (**L**) Probability of wild-type (WT) or LTβRΔ mouse survival after B-ALL transplantation (n=8 mice/group). Mice were transplanted with 3×10⁶ BCR-ABL-expressing B-ALL cells and analyzed on day 8 after transplantation. (**M**) Histogram of LTβR ligand expression in ALL cells. Green, *Ltb*-sufficient; brown, *Ltb*-deficient. (**N**) *Il7*-GFP expression in MSCs. (**O**) Number of non-malignant developing B cell subsets. (**P**) ALL number. (**Q**) Frequency of WT mouse survival after *Ltb*-deficient or *Ltb*-sufficient ALL transplantation (n=7/group). (**R and S**) Effects of *Ltb*-expressing ALLs in MSCs. (**R**) *Ltb⁺/⁺* and *Ltb⁻/⁻* B-ALL numbers in BM 3 weeks after transplantation into *Il7^(GFP/+)* mice. (**S**) *Il7*-GFP expression in MSCs. Gray bar indicates control *Il7^(GFP/+)* mice (no ALL); green bar represents *Il7^(GFP/+)* mice recipient of 3×10⁶ *Ltb⁺/⁺* ALLs; brown bar indicates *Il7^(GFP/+)* mice recipient of 3×10⁶ *Ltb⁻/⁻* ALLs; red bars depict *Il7^(GFP/+)* mice recipient of 9×10⁶ *Ltb⁻/⁻* ALLs (3×*Ltb⁻/⁻*). Bars indicate mean, circles depict individual mice. Data in all panels are representative of two independent experiments. *p<0.05; **p<0.005; ***p<0.0005 unpaired, two-sided, Student's *t* test. ALL, acute lymphoblastic leukemia.

The online version of this article includes the following figure supplement(s) for figure 4:

**Figure supplement 1.** Effects of IL7 administration in acute lymphoblastic leukemia (ALL) and non-leukemic lymphopoiesis in vivo.

ALLs (*Figure 4R and S*). Combined, these studies show that the ALL-induced IL7 downregulation that we reported in previous studies (*Fistonich et al., 2018*; *Zehentmeier and Pereira, 2019*) is mediated by direct delivery of lymphotoxin ligands to LTβR expressed on BM MSCs.

To test if increased lymphopoiesis due to excess IL7 is directly responsible for reduced ALL growth in vivo, we treated mice transplanted with B-ALL cells with recombinant IL7 complexed with a neutralizing anti-IL7 (aIL7, clone M25) monoclonal antibody (*Figure 4—figure supplement 1A*), which increases the half-life of recombinant IL-7 in vivo (*Boyman et al., 2008*). Indeed, mice treated with IL7/aIL7 had significantly higher numbers of developing B cell subsets in the BM (*Figure 4—figure supplement 1B*), particularly of IL7-dependent proB and pre-B cells. Conversely, ALL numbers in BM were significantly reduced, which reflected in significant reductions in peripheral blood and spleen (*Figure 4—figure supplement 1C*). Combined, these data demonstrate that the lymphotoxin-mediated attenuation of IL7 production reduces lymphopoiesis, which results in accelerated ALL growth.

Sleckman and colleagues used an elegant mouse model consisting of Artemis-deficient mice crossed with immunoglobulin heavy chain transgenic mice (Vh147) and with EµBCL2 transgenic mice to study the impact of double stranded DNA break (DSB) repair pathway in B cell development. In these studies, they reported NFκB activation in pre-BCR-expressing pre-B cells that could not repair DSBs due to deficiency in Artemis (*Bredemeyer et al., 2008*; *Bednarski et al., 2016*). In previous studies using this model, we showed that Artemis-deficient pre-B cells could also induce IL7-downregulation in MSCs (*Fistonich et al., 2018*; *Zehentmeier and Pereira, 2019*), suggesting that the LTβR pathway may also be engaged in pre-leukemic states. Consistent with this possibility, when comparing the transcriptome of Artemis-deficient ($Dclre1c^{-/-}$) and Rag1-deficient ($Rag1^{-/-}$) pre-B cells, we noted that Artemis-deficient pre-B cells also expressed significantly higher amounts of LTα and LTβ transcripts (*Bredemeyer et al., 2008*; *Bednarski et al., 2016*). In agreement with these observations, we detected higher amounts of LTα and LTβ protein on the cell surface of $Dclre1c^{-/-}$ pre-B cells than on $Rag1^{-/-}$ pre-B cells (*Figure 5—figure supplement 1A*). Transplantation of $Dclre1c^{-/-}$ BM cells into lethally irradiated $Il7^{GFP/+}$ LTβRΔ mice or control littermate revealed LTβR-dependent $Il7$ downregulation (*Figure 5—figure supplement 1B*), which resulted in a trend towards increased numbers of Artemis-deficient B cells (*Figure 5—figure supplement 1C*). In contrast, Artemis deficiency did not impact myeloid-erythroid production (*Figure 5—figure supplement 1D and E*). Artemis deficiency renders cells unable to repair DSBs which causes a supraphysiological activation of the DNA damage response pathway (*Bednarski and Sleckman, 2012*). To test if the DNA damage response controls LTα and LTβ expression, we treated ALL cells with Etoposide, a chemotherapeutic agent that prevents DSB repair. ALLs upregulated LTα on the cell surface in an Etoposide dose-response manner (*Figure 5—figure supplement 1F and G*). The DNA damage response pathway signals activation of NFκB. Interestingly, treatment with IMD0354 (a small molecule inhibitor of IKKβ-mediated IκB phosphorylation) significantly reduced lymphotoxin α1β2 expression in Etoposide-treated ALLs (*Figure 5—figure supplement 1H*). Combined, these studies demonstrate that LTα and LTβ expression can be activated by DNA damage response pathway.

## CXCR4 signaling potentiates ALL lethality

Prior studies have shown that Gαi-protein coupled receptor signaling in B-lineage cells promotes lymphotoxin α1β2 expression (*Ansel et al., 2000*). Engagement of LTβR expressed on secondary lymphoid organ stromal cells increases the production of B cell chemokines, which further increases lymphotoxin α1β2 expression in B cells, thus establishing a feedforward loop (*Ansel et al., 2000*; *Cyster et al., 2000*). To test if CXCR4 signaling in ALLs promotes lymphotoxin α1β2 expression, we treated ALLs in vitro with a range of CXCL12 concentrations and measured surface LTα. Indeed, ALLs upregulated LTα after exposure to CXCL12 (*Figure 5A*). Furthermore, LTα expression was further increased in ALLs treated with CXCL12 and Etoposide (*Figure 5B*). Although prior studies have also shown that IL7R signaling can promote lymphotoxin α1β2 expression in lymphoid tissue inducer cells (*Yoshida et al., 2002*), in ALL cells IL7R signaling was not sufficient for upregulating surface LTα even at high IL7 concentrations, even though it promoted IL7Ra internalization (*Figure 5—figure supplement 2A and B*). To test if CXCR4 signaling is required for lymphotoxin α1β2 expression in ALLs in vivo, we transplanted $3\times10^6$ ALL cells into WT mice for 1 week and treated them with AMD3100, or saline, 12 hr prior to sacrifice. Indeed, lymphotoxin α1β2 expression was significantly reduced in

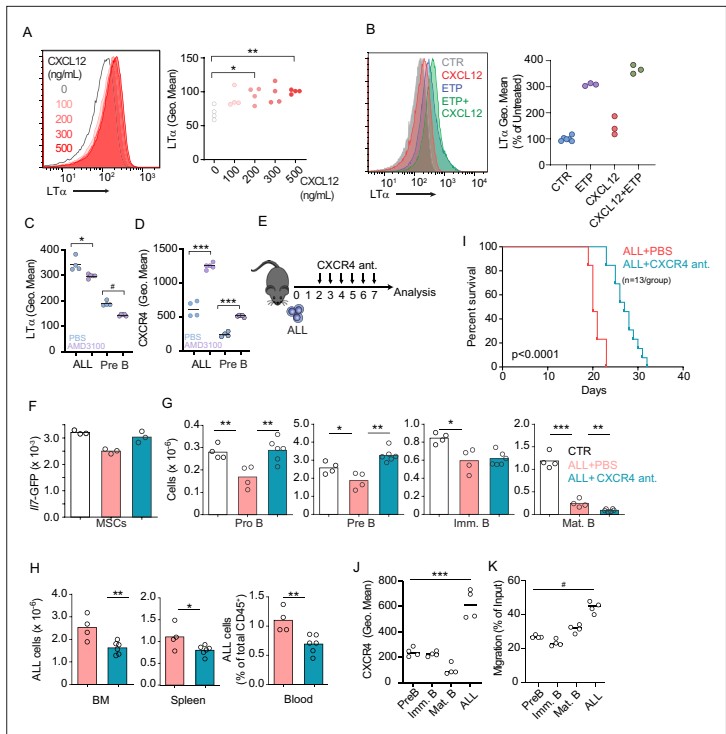

**Figure 5.** CXCR4 signaling and its impact on acute lymphoblastic leukemia (ALL) growth in vivo. (**A**) Histograms of LTα expression in B-ALL cells treated for 16 hr with CXCL12 at the indicated concentrations in vitro. (**B**) Histograms of LTα expression and geometric mean intensity (plotted as percent of untreated cells) in B-ALL cells treated with Etoposide (ETP, 1 µM) alone or in combination with CXCL12 (500 ng/mL) in vitro. (**C and D**) LTα (**C**) and CXCR4 expression (**D**) in ALLs and non-leukemic pre-B cells in the bone marrow (BM) of mice transplanted with 3×10⁶ Ltb⁺/⁺ BCR ABL ALLs. ALLs were allowed to expand in vivo for 7 days. Mice were treated with saline or AMD3100 (80 µg/mouse/i.v.) 18 hr prior to sacrifice. (**E**) Experimental design of data described in panels F–H. Mice were transplanted with 3×10⁶ BCR-ABL-expressing B-ALL cells and treated daily with CXCR4 antagonist starting on day 2 and until day 7; mice were analyzed on day 8. (**F**) *Il7*-GFP expression in mesenchymal stem cells (MSCs). (**G**) Number of non-malignant developing B cell subsets. (**H**) Total ALL number in BM (left), spleen (middle), and B-ALL percentage in peripheral blood (right). (**I**) Frequency of mouse survival after B-ALL transplantation into mice treated with vehicle or CXCR4 antagonist (n=13/group). (**J**) CXCR4 expression on developing B cells and ALLs in BM. (**K**) In vitro chemotaxis of developing B cells and ALLs. Data in panels J and K are from BM of mice 1 week after ALL transplantation. Data are representative of two independent experiments. *p<0.05; **p<0.005; ***p<0.0005 unpaired, two-sided, Student's *t* test. #p<0.05 Mann–Whitney test.

The online version of this article includes the following figure supplement(s) for figure 5:

**Figure supplement 1.** Regulation of Lymphotoxin α1β2 expression by the DNA damage response pathway.

**Figure supplement 2.** Effects of IL7 on lymphotoxin α1β2 expression.

ALLs and in non-leukemic pre-B cells of AMD3100-treated mice (*Figure 5C*). Conversely, AMD3100 increased CXCR4 surface expression in ALL and non-leukemic pre-B cells (*Figure 5D*), as expected (*Broxmeyer et al., 2005*; *Beck et al., 2014*). AMD3100 has a short half-life in vivo thus making it unsuitable for long-term treatment in vivo (*Beck et al., 2014*; *Hendrix et al., 2000*). To test if CXCR4 antagonism also prevents ALL-induced IL7 downregulation in BM MSCs, we transferred 3×10⁶ ALL cells into *Il7*^GFP/+ mice and treated them with an orally bioavailable CXCR4 antagonist (*Dale et al., 2020*) or with vehicle by daily gavage (*Figure 5E*). While control-treated mice showed ALL-induced *Il7*-GFP downregulation in MSCs, mice treated with CXCR4 antagonist maintained IL7 expression within the normal range of mice without ALL (*Figure 5F*). Similarly, developing B cells were significantly reduced in control-treated mice, but their numbers were normal in CXCR4 antagonist treated mice (*Figure 5G*). In contrast, ALL numbers were significantly decreased in the BM and periphery of mice treated with CXCR4 antagonist (*Figure 5H*), which correlated with extended mouse survival (*Figure 5I*).

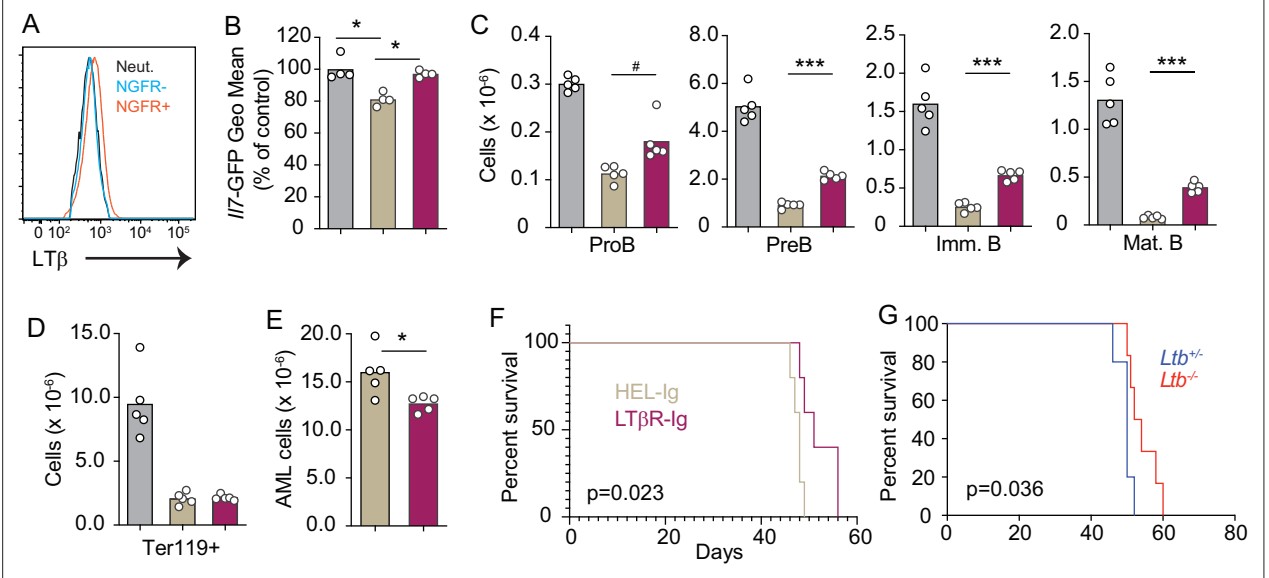

**Figure 6.** Lymphotoxin α1β2 expression in acute myeloblastic leukemias (AMLs) and therapeutic effect of lymphotoxin beta receptor (LTβR) blocking. (**A**) Histograms of LTβ expression in AML cells (NGFR+, red), in non-malignant myeloid cells (NGFR-, blue) and in non-malignant neutrophils (Net., black). (**B**) *Il7*-GFP expression in mesenchymal stem cells (MSCs). (**C**) Number of non-malignant developing B cell subsets in bone marrow. (**D**) Ter119+ red blood cell number. (**E**) AML cell number in bone marrow. (**F**) Probability of mouse survival after AML transplantation following pre-treatment with either HEL-Ig or LTβR-Ig (n=5 per group). (**G**) Probability of mouse survival after *Ltb*-deficient or sufficient AML transplantation (n=10 per group). In panels B–G, comparisons between control (no AML, gray), AML treated with Hel-Ig (peach gray), and AML treated with LTβR-Ig (wine red). Bars indicate mean, circles depict individual mice. Data are representative of two independent experiments. *p<0.05; ***p<0.0005 unpaired, two-sided, Student's *t* test. #p<0.05 Mann–Whitney test.

The online version of this article includes the following figure supplement(s) for figure 6:

**Figure supplement 1.** Kinetics of acute myeloblastic leukemia (AML) growth and disruption of non-malignant hematopoiesis.

Several studies have demonstrated that CXCR4 promotes ALL homing and retention in the BM (*Juarez et al., 2007*; *Sipkins et al., 2005*). Paradoxically, our studies revealed that ALLs induce a small, but significant, downregulation of CXCL12 expression in vivo (*Figure 3D*). However, when comparing CXCR4 protein levels between ALLs and non-leukemic B cell progenitors developing in the same microenvironment, we noted that ALLs express significantly higher amounts of CXCR4 in vivo (*Figure 5J*) and migrate towards a CXCL12 gradient in vitro significantly more than non-leukemic developing B cell subsets (*Figure 5K*). These observations suggest that ALLs are retained more efficiently in the BM than non-leukemic B cell progenitors.

## LTβR signaling promotes AML growth and lethality

As previously mentioned, AML also induces the downregulation of multiple hematopoietic cytokines expressed by MSCs, including IL7 (*Baryawno et al., 2019*). Furthermore, we also found an inverse correlation between *LTB* transcript abundance and patient outcome in a cohort of AML patients from the Cancer Genome Atlas (*Figure 6—figure supplement 1A*). To study the impact of AML growth in non-malignant hematopoiesis, we used a doxycycline (DOX)-inducible mouse model of the MLL-AF9 oncogenic fusion driving AML (*Chen et al., 2019*). In a mixed BM chimera of DOX-induced MLL-AF9 transgenic and competitor WT cells expressing CD45.1, MLL-AF9 expression increased the frequency of AML cells and of myeloid cell subsets expressing NGFR, a reporter for MLL-AF9 (*Figure 6—figure supplement 1B and C*). In contrast, both erythroid and lymphoid subsets were significantly reduced 2 and 4 weeks after MLL-AF9 induction (*Figure 6—figure supplement 1D and E*). Next, we tested if the LTβR pathway is also engaged by AMLs. LTβ expression is higher on MLL-AF9-positive than on MLL-AF9-negative myeloid cells and neutrophils (*Figure 6A*). MLL-AF9-expressing cells induced IL7 downregulation in MSCs, which could be blocked with LTβR-Ig but not with Hel-Ig treatment (*Figure 6B*). The effect of LTβR-Ig treatment also restored B lymphopoiesis partially (*Figure 6C*), but not erythropoiesis (*Figure 6D*), which correlated with reduced AML growth in vivo and extended mouse survival

(*Figure 6E and F*). Similar findings were obtained with AML cells deficient in *Ltb* (*Figure 6G*). Overall, these studies collectively demonstrate that acute lymphoid and myeloid leukemias turn off lymphopoiesis by expressing LTβR ligands and downregulating IL7 production.

## Discussion

Like normal hematopoietic progenitors, leukemia cells physically interact with the BM niche and deliver signals capable of re-programming MSCs and ECs and altering the hematopoietic output (*Baryawno et al., 2019*; *Tikhonova et al., 2019*; *Fistonich et al., 2018*; *Zehentmeier and Pereira, 2019*; *Méndez-Ferrer et al., 2020*). In this study, we showed that lymphotoxin α1β2 delivered by ALL and AML cells to LTβR expressed on MSCs is one mechanism by which leukemic cells dysregulate MSCs and alter hematopoiesis. The fact that AML and ALL cells physically interact with MSCs provides opportunities for LTβR engagement via membrane-bound lymphotoxin ligands.

Previously, we uncovered cell circuits between Artemis-deficient pre-B cells and IL7-producing MSCs that resulted in IL7 downregulation (*Fistonich et al., 2018*; *Zehentmeier and Pereira, 2019*). Here, we demonstrated that the Artemis-deficient pre-B cells express elevated lymphotoxin α1β2 and that IL7 downregulation was dependent on LTβR signaling in MSCs. These findings identify the DSB sensing and repair pathway as one mechanism promoting lymphotoxin α1β2 expression in pre-leukemic cells through the activation of NFκB transcriptional activity. Because upregulation of lymphotoxin α1β2 in pre-B cells carrying unrepaired DSBs promotes IL7 downregulation, we speculate that this mechanism may contribute to the efficient elimination of pre-leukemic B cell progenitors in vivo. Whether the same mechanism is responsible for lymphotoxin α1β2 upregulation in AML cells remains to be determined.

Besides the DSB pathway, chemokine receptor signaling, particularly CXCR5 and CXCR4, have also been shown to promote lymphotoxin α1β2 expression in B-lineage cells (*Ansel et al., 2000*). Our studies revealed that CXCR4 signaling also induces lymphotoxin α1β2 expression in ALL cells in vitro and in vivo. CXCL12 is the most abundant chemokine expressed by MSCs and ECs in BM, and CXCR4 expression levels in AML and ALL cells inversely correlate with patient outcome (*Cancilla et al., 2020*). We suggest that CXCR4 signaling not only enables leukemic cell interactions and delivery of lymphotoxin α1β2 to MSCs but also promote further lymphotoxin α1β2 expression in leukemic cells in a feed-forward loop that shuts down IL7 expression and blocks lymphopoiesis. The fact that our studies also revealed an inverse correlation between lymphotoxin α1β2 expression and ALL and AML patient outcome, as has been described with CXCR4 (*Cancilla et al., 2020*), is in agreement with the data presented in this study showing that CXCR4 and LTβR act in the same axis.

Prior studies using mouse models of AML have shown that CXCL12 is downregulated in MSCs when AML develops (*Baryawno et al., 2019*). Here, we showed that ALL cells induce a small but significant downregulation of CXCL12 expression in MSCs that is partly dependent on LTβR signaling. However, despite the CXCL12 downregulation, ALL cells are still retained in the BM presumably due to higher CXCR4 expression when compared to that observed in non-leukemic large cycling pre-B cells. We speculate that high CXCR4 levels confer competitive advantage over non-leukemic hematopoietic progenitors in BM retention and access to supportive microenvironments for cell growth.

Studies using cell-based therapies with T cells expressing anti-CD19 chimeric antigen receptors (CAR-T cells) have shown promising therapeutic effects against B cell malignancies (*Kochenderfer et al., 2010*; *Kalos et al., 2011*; *Porter et al., 2011*). Although remission has been achieved in some patients (*Grupp et al., 2013*; *Maude et al., 2014*; *Porter et al., 2015*), many patients still relapse due to a variety of reasons including defects in long-term persistence of CAR-T cells (*Shah and Fry, 2019*). A recent study, however, showed that CAR-T engineered to express IL7 improved their long-term persistence (*Adachi et al., 2018*). It is possible that LTβR blocking in combination with CAR-T cell therapy particularly for B cell leukemias may also improve CAR-T cell persistence by increasing IL7 production in BM while not affecting CXCL12 expression.

In this study, we showed that ALL cells enforce the downregulation of several myeloid and lymphoid cytokines produced by MSCs and ECs. These findings are reminiscent of prior observations made with mouse models of AML (*Baryawno et al., 2019*; *Hanoun et al., 2014*). Of all hematopoietic cytokines downregulated by leukemic cells, only IL7 (and partly CXCL12) are regulated by LTβR signaling. Other cytokines, such as SCF (encoded by *Kitl*), M-CSF (encoded by *Csf1*), IL34, FLT3L, etc., are also downregulated but in an LTβR-independent manner. Of note, SCF is a critical cytokine for the survival and

expansion of myelo-erythroid lineage progenitors (*Miao et al., 2020*; *Broudy et al., 1995*; *Broudy, 1997*; *Munugalavadla and Kapur, 2005*), and its consumption in vivo is under competition between hematopoietic stem and progenitor cells (*Miao et al., 2022*). The fact that SCF production is insensitive to LTβR signaling may contribute to explain why myelopoiesis and erythropoiesis are not restored when LTβR signaling is blocked. These findings also raise the possibility that additional receptor(s)/ligand(s) interaction(s) are responsible for controlling the production of other hematopoietic cytokines. Further studies are needed to identify additional pathways responsible for MSC and EC re-programming in response to leukemias.

## Methods

### Mice

C57BL/6NCR (strain code 556, CD45.2+) and B6-Ly5.1/Cr (stain code 564, CD45.1+) were purchased from Charles River Laboratories. *Lepr*-cre mice were purchased from The Jackson Laboratories. *Il7*$^{GFP/+}$ mice were from our internal colony. *Ltb*$^{fl/fl}$ (*Tumanov et al., 2002*) and *Ltbr*$^{fl/fl}$ (*Wang et al., 2010*) mice were bred at Yale Animal Resources Center. DOX-inducible MLL-AF9 (*Hprt*$^{MLL-AF9}$, *Rosa26*$^{rtTA/rtTA}$) transgenic mice were bred at Yale University. Male and female adult mice (8–12 weeks) were used for all experiments. All mice were maintained under specific pathogen-free conditions at the Yale Animal Resources Center and were used according to the protocol approved by the Yale University Institutional Animal Care and Use Committee (2022-11377).

### Adoptive transfer of BCR-ABL-expressing B-ALL cells and in vivo cytokine/cytokine receptor blocking

BCR-ABL-expressing B-ALL cells are developmentally arrested at the pre-B cell stage (kindly provided by Hilde Schjerven, UCSF). B-ALL cells were injected into recipient mice by tail vein, and then analyzed at different time points. For cytokine/cytokine receptor blocking, 150 µg of LTβR-Ig/HEL-Ig (Biogen), anti-TNF antibody (Bio-X-Cell #BE0058), anti-mouse IL1b (Bio-X-Cell #BE0246), anti-mouse IFNAR1 (Bio-X-Cell #BE0241), or anti-mouse IFNγ (Bio-X-Cell #BE BE0055) were injected intravenously via retro-orbital sinus immediately prior to ALL cell injection (tail vein). Then antibody treatment was administered every 5 days with same amount.

### Flow cytometry

BM MSCs were isolated as previously described (*Cordeiro Gomes et al., 2016*; *Zehentmeier et al., 2022*; *Lim et al., 2023*). Briefly, long bones were flushed with HBSS supplemented with 2% of heat-inactivated fetal bovine serum, penicillin/streptomycin, L-glutamine, HEPES, and 200 U/mL Collagenase IV (Worthington Biochemical Corporation) and digested for 30 min at 37°C. Cells were filtered through a 100 µm nylon mesh and washed with HBSS/2% FBS. All centrifugation steps were done at 1200 rpm for 5 min and all stains were done on ice. LEPR stains were done for 1 hr and all other stains for 20 min on ice. BM MSCs were identified as CD45⁻ Ter119⁻ CD31⁻ CD144⁻ LEPR⁺ cells. For analysis of hematopoietic populations, long bones were flushed with DMEM supplemented with 2% fetal calf serum, penicillin/streptomycin, L-glutamine, and HEPES. Red blood cells were lysed with ammonium chloride buffer.

Hematopoietic cell populations were identified as follows: proB: CD19⁺ CD93⁺ IgM⁻ cKit⁺; pre-B: CD19⁺ CD93⁺ IgM⁻ cKit⁻; immature B: CD19⁺ IgM⁺ CD93⁺; mature B: CD19⁺ IgM⁺ CD93⁻; immature neutrophils: CD115⁻ Gr1⁺ CD11b$^{hi}$ CXCR4$^{hi}$; mature neutrophils: CD115⁻ Gr1$^{hi}$ CD11b$^{lo}$; immature monocytes: CD115⁺ Gr1⁺ CXCR4$^{hi}$; mature monocytes: CD115⁺ Gr1⁺ CXCR4$^{lo}$; GMP: Lineage⁻ cKit⁺ SCA-1⁻ CD34⁺ CD16/32$^{hi}$; immature and mature erythrocytes: Ter119⁺ CD71⁻ (mature) or CD71⁺ (mature). The lineage cocktail was as follows: CD19, B220, CD3e, CD4, Gr1, NK1.1, Ter119, CD11b, and CD11c.

Measurements of LTα and LTβ expression were performed using anti-mouse LTα and LTβ antibodies (a gift from Biogen). A list of antibodies and conditions used is provided in *Supplementary file 3*.

### Generation of *Ltb*-deficient BCR-ABL B-ALL cells

YFP tagged BCR-ABL plasmid was kindly provided by Dr Hilde Schjerven (UCSF). Pre-B cells were sorted *from Ltb*$^{+/-}$or *Ltb*$^{-/-}$ mouse BM by gating on CD19+CD93+IgM-cKit-. Sorted pre-B cells were

transduced with BCR-ABL YFP retroviruses. After infection, pre-B cells were cultured in DMEM supplemented with 20% FBS, penicillin/streptomycin, L-glutamine, HEPES, 0.05 mM 2-mercaptoethanol, and 100 ng/mL recombinant murine IL7 (peprotech, 217-17). After 2 days of culture, cells were cultured in the same media without IL7 and expanded for transplantation. Recipient mice received 6 Gy gamma irradiation before transplantation.

## Induction of AML in vivo

MLL-AF9 mice were maintained and genotyped as previously described (*Chen et al., 2019*). Two million BM cells recovered from MLL-AF9 transgenic mice were mixed with 1 million BM cells from WT congenic CD45.1 mice; BM mixtures were transplanted into lethally irradiated CD45.1 recipients. Two weeks after transplantation, mice were fed with 1 g/L Dox in the drinking water sweetened with 10 g/L sucrose. For *Ltb*-deficient MLL-AF9 experiment, *Ltb*$^{-/-}$ mice were crossed with MLL-AF9 transgenic mice to generate *Ltb*$^{+/-}$, MLL-AF9 and *Ltb*$^{-/-}$, MLL-AF9 mice.

## Treatments with CXCR4 antagonists

CXCR4 receptor antagonist X4P-X4-185-P1 (X4 Pharmaceuticals) was dissolved in 50 mM citrate buffer, pH 4.0. Mice were treated at 100 mg/kg daily by oral gavage 2 days after ALL adoptive transfer. Control mice were treated by oral gavage of citrate buffer alone. Some experiments were performed with the CXCR4 antagonist AMD3100 (Sigma). Briefly, mice were transplanted with $3 \times 10^6$ ALL cells i.v. (tail vein) and ALLs were allowed to expand in vivo for 7 days. Mice were treated every 6 hr with 80 µg AMD3100 dissolved in saline (200 µL/mouse) i.v. for a period of 18 hr prior to sacrifice.

## IL7 treatment in vivo

Recombinant murine IL7 of 1.5 µg (peprotech, 217-17) was pre-incubated with 15 µg IL7 antibody (clone M25, Bio-X-Cell) for 20 min at room temperature. Mice were treated i.v. with IL7/aIL7 complex on day 6 and seven after adoptive transfer of BCR-ABL-expressing B-ALL cells. Mice were analyzed 48 hr after the last IL7/aIL7 treatment.

## In vitro studies with BCR-ABL-expressing B-ALL cells

### Role of CXCR4 signaling in lymphotoxin α1β2 expression

One million ALL cells were incubated with 100, 200, 300, or 500 ng/mL CXCL12 (R&D, 460-SD-050) for 16 hr, then stained with anti-mouse LTα antibody (a gift from Biogen) and analyzed by flow cytometry.

### Role of DNA damage repair pathway in lymphotoxin α1β2 expression

Etoposide treatment: One million ALL cells were incubated with 0.01, 0.1, 0.5, 1, 2, 5, 15, 50 µM Etoposide for 16 hr before analyzing LTα expression by flow cytometry.

### Role of NFκB signaling in lymphotoxin α1β2 expression

IMD 3504 (Abcam, Ab144823) was dissolved in DMSO, and dilutions prepared in saline. One million ALL cells were incubated with Etoposide for 16 hr in the presence of carrier or IMD 3504 at the indicated concentrations. Cells were stained and analyzed for LTα expression by flow cytometry.

## MSC sorting and RNA-sequencing

BM stromal cells were isolated as previously described (*Cordeiro Gomes et al., 2016*; *Zehentmeier et al., 2022*; *Lim et al., 2023*). Briefly, hematopoietic cells were depleted by staining with biotin-conjugated CD45 and Ter119 antibodies followed by magnetic depletion with Dynabeads Biotin Binder (Invitrogen #11047). Remaining cells were stained with antibodies against CD31, CD144, and PDGFRα, and MSCs were sorted as CD45$^-$ Ter119$^-$ CD31$^-$ CD144$^-$ PDGFRα$^+$ cells using a BD FACS Aria II. Cells were sorted directly into 350 µL RLT plus buffer (Qiagen) and RNA extracted using the RNeasy Plus Micro Kit (Qiagen #74034). RNA-sequencing was performed by the Yale Center for Genome Analysis using the Illumina HiSeq2500 system, with paired-end 75 bp read length. The sequencing reads were aligned onto *Mus musculus* GRCm38/mm10 reference genome, using the HISAT2 software. The mapped reads were converted into the count matrix with default parameters using the StringTie2 software, followed by the variance stabilizing transformation offered by DESeq2. DEGs

were identified using the same software, DESeq2, based on a negative binomial generalized linear models and visualized in hierarchically clustered heatmaps using the pheatmap R package.

### Statistical analyses

We used Student's *t* test to determine if differences between experimental grous with normal distribution were statistically significant. Shapiro–Wilk normality test was performed to assess distribution normality (*Supplementary file 4*). We employed a non-parametric Mann–Whitney test for experimental groups with abnormal distribution. Differences in mouse survival between experimental groups were analyzed by the Kaplan–Meyer method.

### Patient outcome and gene expression microarray data

The B-ALL gene expression microarray and patient outcome data were obtained from the National Cancer Institute TARGET Data Matrix (http://targetnci.nih.gov/dataMatrix/TARGET_DataMatrix.html) of the Children's Oncology Group (COG) Clinical Trial P9906 with the GEO database accession number GSE11877 (*Kang et al., 2010*). The patients were segregated into two groups according whether they had above or below the median expression level of a gene (i.e., the average of multiple probesets for a gene) or above the top 25% or below the bottom 25% expression level of a gene. OS or RFS probabilities were estimated using the Kaplan–Meier method, and log-rank test (two-sided) was used to compare survival differences between the two patient groups. R package 'survival' version 2.35-8 was used for the survival analysis.

## Acknowledgements

We thank Dr Alexei Tumanov (UT San Antonio) for providing *Ltb* and *Ltbr* floxed mouse strains; Dr Hilde Schjerven for providing BCR-ABL1 B-ALL cells and BCR-ABL1 retroviral expression plasmid. We thank Dr Art Taveras (X4 Pharmaceuticals) for providing CXCR4 antagonist. We thank Dr Linda Burkly (Biogen) for providing LTα and LTβ antibodies, LTβR-Ig and Hel-Ig. These studies were funded by the NIH (R01AI113040, R21AI133060, R35CA197628, R01AI164692, and R21AI146648). XF was funded by the NIH (T32 DK007356).

## Additional information

#### Competing interests

Jungmin Choi: Reviewing editor, *eLife*. The other authors declare that no competing interests exist.

#### Funding

| Funder | Grant reference number | Author |
| --- | --- | --- |
| National Institutes of Health | R01AI113040 | Joao Pedro Pereira |
| National Institutes of Health | R21AI133060 | Joao Pedro Pereira |
| National Institutes of Health | R35CA197628 | Marcus Müschen |
| National Institutes of Health | R01AI164692 | Marcus Müschen |
| National Institutes of Health | R21AI146648 | Marcus Müschen |
| National Institutes of Health | T32 DK007356 | Xing Feng |

The funders had no role in study design, data collection and interpretation, or the decision to submit the work for publication.

## Author contributions
Xing Feng, Investigation, Methodology, Writing - review and editing; Ruifeng Sun, Investigation, Methodology; Moonyoung Lee, Software, Formal analysis; Xinyue Chen, Shangqin Guo, Resources, Writing - review and editing; Huimin Geng, Resources, Software; Marcus Müschen, Resources, Software, Writing - review and editing; Jungmin Choi, Resources, Software, Supervision, Writing - review and editing; Joao Pedro Pereira, Conceptualization, Funding acquisition, Investigation, Writing - original draft, Writing - review and editing

## Author ORCIDs
Xing Feng http://orcid.org/0000-0002-2253-4180
Xinyue Chen http://orcid.org/0000-0001-8288-7685
Shangqin Guo http://orcid.org/0000-0003-1157-0423
Jungmin Choi http://orcid.org/0000-0002-8614-0973
Joao Pedro Pereira http://orcid.org/0000-0002-5694-4938

## Ethics
All mice were maintained under specific pathogen-free conditions at the Yale Animal Resources Center and were used according to the protocol approved by the Yale University Institutional Animal Care and Use Committee. (2022-11377).

## Decision letter and Author response
Decision letter https://doi.org/10.7554/eLife.83533.sa1
Author response https://doi.org/10.7554/eLife.83533.sa2

# Additional files

## Supplementary files
• Supplementary file 1. Acute lymphoblastic leukemia (ALL)-induced gene expression changes in Lepr+ mesenchymal stem cells (MSCs). Differentially expressed genes analyzed by bulk RNA-sequencing of Lepr+ MSPCs isolated from resting wild-type (WT) mice or Hel-Ig treated WT mice transplanted with ALL cells for 2 weeks. Related to *Figure 3*.

• Supplementary file 2. Lymphotoxin beta receptor (LTβR)-regulated genes in Lepr+ MSPCs during acute lymphoblastic leukemia (ALL) progression. Differentially expressed genes analyzed by bulk RNA-sequencing of Lepr+ MSPCs isolated from ALL transplanted wild-type mice treated with control or LTβR-Ig (100 μg/mouse/week) for 2 weeks. Related to *Figure 3*.

• Supplementary file 3. List of antibodies used in this study.

• Supplementary file 4. Shapiro–Wilk normality test of the experimental groups examined.

• MDAR checklist

## Data availability
Accession number to RNA expression data were deposited in NCBI (GSE221243).

The following dataset was generated:

| Author(s) | Year | Dataset title | Dataset URL | Database and Identifier |
|---|---|---|---|---|
| Feng X, Sun R, Lee M, Chen X, Guo S, Geng H, Müschen M, Choi J, Pereira JP | 2022 | Cell circuits between leukemic cells and mesenchymal stem cells block lymphopoiesis by activating lymphotoxin-beta receptor signaling | https://www.ncbi.nlm.nih.gov/geo/query/acc.cgi?acc=GSE218505 | NCBI Gene Expression Omnibus, GSE218505 |

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
