## [Editor Report]

This study investigates a novel pathway by which leukemic cells remodel the bone marrow niche to promote their expansion at the expense of normal hematopoiesis. Feng X, Pereira JP et al. convincingly demonstrate a positive feedback loop between leukemic cells and stromal cells mediated by lymphotoxin produced by cancer cells and its receptor expressed by bone marrow stromal cells. The authors provide compelling evidence suggesting that this pathway disrupts normal blood production and provides a competitive advantage to leukemic cells.

---

## [Decision Letter]

**Decision letter after peer review:**

Thank you for submitting your article "Cell circuits between leukemic cells and mesenchymal stem cells block lymphopoiesis by activating lymphotoxin-β receptor signaling" for consideration by *eLife*. Your article has been reviewed by 2 peer reviewers, and the evaluation has been overseen by a Reviewing Editor and Tony Ng as the Senior Editor. The reviewers have opted to remain anonymous.

Essential revisions:

The consensus is that this is a very interesting potential novel axis by which leukemic cells impede normal hematopoiesis. Yet, additional important controls seem to be required to substantiate the pathway at play and exclude other effects (such as homing of leukemic cells, or leukemic cell burden possibly impacting IL7 expression), particularly for those experiments with partial effects, as explained in the recommendations by the reviewers. The unexpected effect of lymphotoxin reducing (instead of inducing) CXCL12, discrepancies among the effects on WT, preleukemic and leukemic cells and different results across figures should be clarified as well, following the reviewers' recommendations. Mislabelled axis with the 10-x format should be corrected as well.

*Reviewer #1 (Recommendations for the authors):*

One weakness of this study is that some of the differences are very small or can vary, warranting cautious interpretation of the results. The effect of LTbR block in Figure 2I appears very small (no statistical test is shown). In Figure 6F and G, the effects were also very small. These results leave it unclear whether the positive feedback loop is pathologically important or not. In Figure 4, the effect of ALL cells on hematopoiesis is very weak in 4C but prominent in 4N. It was not clear from the figure legends whether these experiments carried out in different ways.

Regarding the contribution of CXCR4, it appears to be difficult to conclude that its effect is via an induction of lymphotoxin expression in leukemic cells. Rather it may simply affect stroma interaction.

To further support their conclusions, authors need to examine how DNA damage response induces the expression of lymphotoxin and/or how lymphotoxin receptor signaling suppresses the expression of IL7.

Others

In Supp Figure 5, red and blue bars to show time points are confusing (these are with no manipulations).

*Reviewer #2 (Recommendations for the authors):*

1) Concerning statistics. Student's parametric t-test can only be used if the samples have a normal distribution (tested with a D'Agostino-Pearson or Shapiro-Wilk normality test). When normality is not reach (p<0.05), the authors must perform a non-parametric Mann Whitney test. Of note, normality will never be reached with n<4 samples. Furthermore, for two-tailed non parametric test and an α of 0.05, and according to the Mann-Whitney table, statistical significance cannot be reached for 3 vs. 3 or 4 vs. 3 samples per group. Please follow these rules.

2) p3, in the first paragraph of the second sub-section, the reference Zehentmeier et al., 2022 is not formatted

3) In the last sentence p3, the authors claim that there is a significant extension of mouse survival with the LTbR-Ig treatment. Statistics must be shown in the related Figure 2I.

4) p5, last sentence. I guess the authors meant that ALL numbers were significantly decreased and not increased?

[Editors’ note: further revisions were suggested prior to acceptance, as described below.]

Thank you for resubmitting your work entitled "Cell circuits between leukemic cells and mesenchymal stem cells block lymphopoiesis by activating lymphotoxin-β receptor signaling" for further consideration by *eLife*. Your revised article has been evaluated by Simón Méndez-Ferrer (Reviewing Editor) and Tony Ng (Senior Editor).

The manuscript has been greatly improved but there are some remaining issues that need to be addressed, as outlined below:

– Additional discussion needs to be provided on the interplay between LTab, CXCL12, and IL7 for a more balanced interpretation and consideration of possible alternative explanations for the effects observed.

– The labelling of Figure 2D (LTbR-Ig and HEL-Ig) is inverted compared with the previous version.

– Normality and homoscedasticity tests are missing to support the Gaussian distribution and the use of t-tests.

*Reviewer #2 (Recommendations for the authors):*

The authors clarified why some of the differences were small in their experimental settings. These explanations are reasonable and difficult to address experimentally. However, other results from experiments using sophisticated genetic models, are mostly compelling. In addition, the authors made clear how DNA damage increased the expression of lymphotoxin (revised experiments in Figure 5). Overall, this manuscript is ready for publication. Congratulations on your meticulous and beautiful work.

*Reviewer #3 (Recommendations for the authors):*

Feedback on the manuscript for the authors

It would have been appreciated to have a revised version with track changes to better evaluate the revisions of the manuscript.

Concerning statistics:

The authors cannot take for granted the fact that previous studies mishandled statistics to justify their own mishandling. By doing so, the authors ask the reviewers and the audience of *eLife* to accept the fact that their interpretation could be wrong. A Gaussian distribution is a prerequisite to a t-test and there should be no discussion about that.

The labelling of Figure 2D has been inverted (LTbR-Ig and HEL-Ig).

---

## [Author Response]

Reviewer #1 (Recommendations for the authors):One weakness of this study is that some of the differences are very small or can vary, warranting cautious interpretation of the results. The effect of LTbR block in Figure 2I appears very small (no statistical test is shown). In Figure 6F and G, the effects were also very small. These results leave it unclear whether the positive feedback loop is pathologically important or not. In Figure 4, the effect of ALL cells on hematopoiesis is very weak in 4C but prominent in 4N. It was not clear from the figure legends whether these experiments carried out in different ways.

We thank the reviewer for the valuable criticisms.

The statistical analysis has been included in Figure 2I. The variability in its magnitude is merely technical for the following reasons. in vivo treatment with LTbR-Ig had to be performed under sub-optimal conditions. Ideally we would have preferred to treat mice bi-weekly to ensure full saturation but this was impossible for lack of sufficient antagonist (a generous gift from Biogen). The effects described with LTbR conditional KO mice are also sub-optimal because Lepr-cre can only effectively delete LTbR protein in ~ 80% of MSCs, despite being the best genetic strategy currently available for genomic targeting of MSCs. The result we think reflects the true physiological impact of the LTbR pathway in BCR-ABL ALL is that depicted in Figure 4P with Ltb-deficient ALLs, which shows a significant and more promising impact in ALL growth in vivo. Having said this, we agree with the reviewer that the physiological impact of LTbR signaling in MSCs, even though significant, is still modest and possibly better explored in combination therapy. For example, studies by others have shown a positive effect of IL7 in CAR-T therapy (Adachi et al. Nat. Biotech 2018) against solid tumors and perhaps our study will be useful for future studies examining the therapeutic potential of CAR-T and LTbR targeted therapies against leukemia. We revised the manuscript to include this discussion point.

Regarding the contribution of CXCR4, it appears to be difficult to conclude that its effect is via an induction of lymphotoxin expression in leukemic cells. Rather it may simply affect stroma interaction.

We agree with the reviewer. In fact we apologize for not making it clear that the effect of CXCR4 signaling is: First, by mediating physical interactions between ALL (and non-leukemic leukocytes) and MSCs in the BM. Second, by further promoting Lta1b2 expression in ALLs (as it does on B-lymphocytes, Ansel et al. Nature 2000), it potentiates IL7 downregulation in MSCs. We revised the manuscript to make this point clearer.

To further support their conclusions, authors need to examine how DNA damage response induces the expression of lymphotoxin and/or how lymphotoxin receptor signaling suppresses the expression of IL7.

We thank the reviewer for the valuable suggestion. We’ve spent a considerable amount of time studying how Il7 production is transcriptionally regulated. Unfortunately, we are still a long way from understanding it completely. This question alone is a project in of itself and thus we think it falls outside of the scope of this manuscript. Regarding the mechanisms driving Lymphotoxin a1b2 upregulation in leukemia cells, we hypothesized that by being induced by the DSB pathway (as Sleckman and colleagues have shown) that NFkB might be involved. We tested this possibility using small molecule antagonist of IKKb-mediated IkBa phosphorylation (IMD-0354) and found that it reduces LTa1b2 expression in ALLs treated with the DNA-damaging agent Etoposide. These data have been included in Figure 5—figure supplement 4 of the revised manuscript.

OthersIn Supp Figure 5, red and blue bars to show time points are confusing (these are with no manipulations).

We’ve corrected this figure in the revised manuscript.

Reviewer #2 (Recommendations for the authors):1) Concerning statistics. Student's parametric t-test can only be used if the samples have a normal distribution (tested with a D'Agostino-Pearson or Shapiro-Wilk normality test). When normality is not reach (p<0.05), the authors must perform a non-parametric Mann Whitney test. Of note, normality will never be reached with n<4 samples. Furthermore, for two-tailed non parametric test and an α of 0.05, and according to the Mann-Whitney table, statistical significance cannot be reached for 3 vs. 3 or 4 vs. 3 samples per group. Please follow these rules.

We performed new experiments and revised most figures to included more mice. However, in some cases, we did not change the original figures for ethical reasons. For example, in Figure 1 we show dramatic effects of ALL growth in hematopoietic lineages after 21 days by examining 3 mice. The impact is so large that we consider it unethical to perform additional experiments for a sole statistical purpose. Furthermore, numerous studies have included parametric student’s t test analyses for n<4 samples (including many studies on bone marrow niches; for an example see Fujisaki et al. Nature 2011).

2) p3, in the first paragraph of the second sub-section, the reference Zehentmeier et al., 2022 is not formatted

This has been corrected.

3) In the last sentence p3, the authors claim that there is a significant extension of mouse survival with the LTbR-Ig treatment. Statistics must be shown in the related Figure 2I.

This has been included.

4) p5, last sentence. I guess the authors meant that ALL numbers were significantly decreased and not increased?

This has been corrected.

[Editors’ note: further revisions were suggested prior to acceptance, as described below.]

The manuscript has been greatly improved but there are some remaining issues that need to be addressed, as outlined below:– Additional discussion needs to be provided on the interplay between LTab, CXCL12, and IL7 for a more balanced interpretation and consideration of possible alternative explanations for the effects observed.

In the revised manuscript we now include two new panels in Figure 5 (J and K) that show increased CXCR4 expression and CXCR4-mediated migration to CXCL12 in ALLs when compared to either small or large preB cells. We believe these data lend support to a model where ALLs are more competent than non-leukemic B cell progenitors in homing and retention in BM even when its ligand CXCL12 expression is slightly downregulated.

– The labelling of Figure 2D (LTbR-Ig and HEL-Ig) is inverted compared with the previous version.

We apologize for the oversight; this has been corrected.

– Normality and homoscedasticity tests are missing to support the Gaussian distribution and the use of t-tests.

A new table 4 has been included. This table contains Normality and homoscedasticity tests for all experimental groups.

Reviewer #3 (Recommendations for the authors):Feedback on the manuscript for the authorsIt would have been appreciated to have a revised version with track changes to better evaluate the revisions of the manuscript.Concerning statistics:The authors cannot take for granted the fact that previous studies mishandled statistics to justify their own mishandling. By doing so, the authors ask the reviewers and the audience of eLife to accept the fact that their interpretation could be wrong. A Gaussian distribution is a prerequisite to a t-test and there should be no discussion about that.

A new table 4 includes normality tests for all experimental groups examined in this study.

The labelling of Figure 2D has been inverted (LTbR-Ig and HEL-Ig).

This has been corrected.